# Evolution of Global Food Trade Patterns and Its Implications for Food Security Based on Complex Network Analysis

**DOI:** 10.3390/foods10112657

**Published:** 2021-11-02

**Authors:** Jieyong Wang, Chun Dai

**Affiliations:** 1Institute of Geographic Sciences and Natural Resources Research, Chinese Academy of Sciences, Beijing 100101, China; daic.19s@igsnrr.ac.cn; 2Key Laboratory of Regional Sustainable Development Modeling, Chinese Academy of Sciences, Beijing 100101, China; 3College of Resources and Environment, University of Chinese Academy of Sciences, Beijing 100049, China

**Keywords:** food security, global food trade, complex network analysis, evolution

## Abstract

Global food trade is an integral part of the food system, and plays an important role in food security. Based on complex network analyses, this paper analyzed the global food trade network (FTN) and its evolution from 1992 to 2018. The results show that: (1) food trade relations have increased and global FTN is increasingly complex, efficient, and tighter. (2) Global food trade communities have become more stable and the trade network has evolved from “unipolar” to “multipolar”. (3) Over the nearly 30-year period, the core exporting countries have been stable and concentrated, while the core importing countries are relatively dispersed. The increasingly complex food trade network improves food availability and nutritional diversity; however, the food trade system, led by several large countries, has increased the vulnerability of some countries’ food systems and brings about unsafe factors, such as global natural disasters and political instability. It is supposed to establish a food security community to protect the global food trade market, address multiple risks, and promote global food security.

## 1. Introduction

Food security will be a worldwide challenge in the coming decades [1,2]. The global food supply has been under the pressure of increased demand, due to a growing population, dietary changes, and increasing biofuel needs [3,4]. In less developed regions, there are still many people struggling with starvation while in regions with better economic development, people pursue higher-quality food consumption [5,6]. Although food production has achieved remarkable results, food supply has some uncertainty because of the growing competition for land and water resources, as well as the impact of global climate change [2,7,8]. The COVID-19 pandemic has exposed the fragility of global food supply and raised the specter of a global food crisis [9,10].

With the deepening development of globalization, the importance of global food trade to food security has been widely recognized [11]. More and more countries have changed from being self-sufficient to participating in the global food trade [12,13]. Geographical barriers have ceased to be obstacles due to the development of transportation and fresh-keeping technologies in past years. As the resource endowment and the levels of food production and consumption are different among countries, global trade has become a great way to adjust the disconnect between supply and demand and contribute to the sustainable utilization of resources [14,15]. However, global food trade can also expose countries to external supply risks due to the increased interdependence of food security among countries [16]. Trade entities and their relationships have formed a global food trade system; thus, it is helpful to clarify the status and interdependence of different countries in this global trade and to identify the source of external supply risks from the perspective of a cereal trade network.

Analyses of global food trade patterns and evolution has been the subject of a number of studies. These have examined the underlying structure of food trade relationships to understand the potential risks of food security [17,18,19,20]. Current studies mainly focus on assessing the stability and resilience of the food trade system under different conditions [21,22,23,24] and have also observed shock transmission and dynamic predictions of resilience to shock so as to prepare for better responses in the future [25]. In addition, some researchers used complex networks to analyze specific issues. One such topic is exploring the features of the global food trade from the perspective of competition, while the other is from the perspective of contaminant tracing and similarities in food safety regulations [17,26]. Additionally, complex networks have been applied to analyze the structure of global virtual water trade, which is associated with the international food trade. These studies are of great significance for improving water resource utilization efficiency and achieving food security [27,28]. Based on a review of the existing literature, we found that the pattern and evolution of the global cereal trade system, focusing on three staple crops simultaneously, received little attention. International scholars have performed in-depth analyses of factors affecting food security, including the role of the global food trade [11]. In this study, we discuss food trade patterns and changes from the perspective of a complex network. The complex global food trade can be abstracted as a network structure. Every country in the network is affected by the others and unstable factors in the network will also become potential risks to food security [21].

Wheat, rice, and maize are the most important cereal crops in the context of population diets worldwide. They are widely produced, consumed, and traded across the world. As a dietary staple, there are large import or export volumes of wheat, rice, and maize every year. Measured in terms of caloric content, it was calculated in Paolo’s study that they make up nearly 50% of the global diet. Thus, here, we selected the trade flow of wheat, rice, and maize as being representative of the cereal trade.

The goal of this study is to explore the patterns and evolutions of the global cereal trade. We address following questions. What kind of network is it? What are the spatial patterns of the global food trade? How did the patterns change? Do these countries tend to trade in clusters? Which countries play a leading role? This paper includes three main parts. First, we construct a global food trade network and reveal its characteristics. Second, we explore the community’s structure and its evolution. Third, we identify the core actors whose behaviors have a strong influence on the trade network.

## 2. Materials and Methods

### 2.1. Data Sources and Processing

Bilateral trade data was downloaded from the Statistics Division of the Food and Agriculture Organization (FAOSTAT) [29]. In 1992, the Soviet Union dissolved and the Cold War ended. China established a socialist market economy in that year and further integrated itself into the world trading system. The world’s political and economic patterns changed significantly in 1992. Thus, we selected the annual import quantity data of cereal commodities for 190 countries, from 1992 to 2018. Without considering the price factor, we selected trade quantities instead of trade volumes. The names of the commodities are wheat, rice paddy, and maize. These are raw grains, excluding manufactured grain products. In order to obtain the total cereal trade relationship matrix, we aggregated the trade quantities of wheat, rice, and maize.

### 2.2. Network Modeling

A complex network model can clearly observe the interactions between nodes and the structural characteristics of the cooperative network [30]. The global food trade network is a complex system composed of “nodes (trading countries/regions)”, “edges (trade links)” and “edge weights (bilateral trade volume)”, which shows the characteristics of a complex network [31]. Hence, the complex network model is conducive to the visual representation of multiple actors in food trade patterns worldwide, which can provide insights into the underlying structures of trade flow, identify key factors within the network of which behaviors may have a strong influence on the remainder of the network, and judge other descriptive factors.

In this case, the global trade network is constructed by taking food trading countries as nodes and food trade links between countries as edges, introducing trade flows to represent nodes. A global *FTN* can be abstracted as a weighted complex network; *G* = (*N*, *E*, *W*). *N* is a node set composed of food trading countries; *E* is the edge set of food trade relations between countries; *W* is the set of functions of the trade quantity relationship between two countries; and $w_{ij}\left(t\right)$ denotes the trade quantity between node i and node j, i.e., the weight of edges. When $w_{ij}\left(t\right)=0$, it means that there is no food trade relationship between nation i and nation j, and when $w_{ij}\left(t\right)>0$, it means that there are trade flows between nation i and nation j. The network can be expressed by a topological adjacency matrix, *A* = $\left(a_{ij}\right)$, i.e.,
(1)$$a_{ij}\left(t\right)=\{\begin{array}{c} 1\;w_{ij}\left(t\right)>0\\ 0\;w_{ij}\left(t\right)=0 \end{array}$$

The network structure of each year constitutes a network snapshot. It is necessary to select appropriate metrics to study the characteristics and evolution of the network structure.

### 2.3. Network Measures

In this study, we used degree and weighted degree to analyze the links between countries and their positions in the network. Density (*d*), average clustering coefficient (*ACC*) and average path length (*APL*) were used to depict the overall feature of the global *FTN*.

Degree (*K*) Node degree represents the number of countries that have direct trade correlations with node $K_{i}$, which is defined as $K_{i}\left(t\right)=\sum_{j=1}^{N}a_{ij}$. In directed networks, it can be divided into out-degree and in-degree, which are defined as $K_{i}^{out}=\sum_{j=1}^{N}a_{ij}$ and $K_{i}^{in}=\sum_{j=1}^{N}a_{ji}$, respectively. $a_{ij}$ and $a_{ji}$ represent the outflow and inflow of food trade, respectively. N represents the total number of trading countries. The higher the degree value, the more countries have trade relations with a country and greater that country’s influence in the trade network.

Weighted degree (*W*) Weighted degree refers to the trade quantity between two countries. In this work, it can be divided into weighted out-degree and weighted in-degree, which are defined as $W_{i}^{out}=\sum_{j=1}^{N}a_{ij}w_{ij}$ and $W_{i}^{out}=\sum_{j=1}^{N}a_{ji}w_{ji}$, respectively. $w_{ij}$ and $w_{ji}$ represent the export trade quantity and import trade quantity of two countries with trade links, respectively. The higher the value, the greater the volume of trade between two countries.

Density (*d*) The density of the network refers to the proportion of actual trade links to all possible trade links, which are defined as $d=\frac{2e}{n\left(n-1\right)}$, and the range of the value is [0, 1]. It is used to measure the overall tightness among countries participating in international trade. A high density means a tight network; where e is the number of connections that actually exist and n represents the number of nodes.

Average clustering coefficient (*ACC*) The clustering coefficient refers to the possibility of whether a country has trade relations among its partners. The average clustering coefficient reflects the average clustering degree around the whole network, which is defined as $ACC=\frac{1}{n}\sum_{i=1}^{n}\frac{e_{i}}{k_{i}\left(k_{i}-1\right)}$. Where $k_{i}$ represents the node degree of node i and $e_{i}$ denotes the number of edges between $k_{i}$ neighbors of i.

Average path length (*APL*) The average path length refers to the average value of the shortest paths between all node pairs in the network, which is defined as $APL=\frac{1}{n\left(n-1\right)}\sum_{i}\sum_{j}d\left(i,j\right)$. d(i,j) represents the shortest path between node i and node j in the network. It is an index to measure the trade transmission efficiency of the network. The shorter the *APL*, the greater the transport efficiency of the network.

### 2.4. Community Discovery

Communities in complex networks refer to subsets of network nodes. Countries in the whole network can be divided into several communities. The community discovery method is used to identify the internal structure of the trade network, based on the actual trade flows between countries [32]. In the specific calculation process, a modular optimization algorithm is used. Modularity as an index is used to measure the quality of the partition obtained by the algorithm. The modularity of a partition, which has a scalar value between −1 and 1, evaluates the density of links inside communities compared to links between communities. The modularity of a partition (*Q*) can be defined as:(2)$$Q=\frac{1}{2m}\underset{ij}{\sum }\left[w_{ij}-\frac{A_{i}A_{j}}{2m}\right]\delta \left(c_{i},c_{j}\right)$$
where $w_{ij}$ is the weight of the edge between nodes i and j. In the unweighted network model, when a trade relation exists between two countries, the value will be 1; otherwise, it will be 0. $A_{i}$ and $A_{j}$ represent the sum of all trade quantities of nodes i and j respectively. *m* is the total trade quantities of the entire network. $\delta \left(c_{i},c_{j}\right)$ indicates that whether the two nodes belong to the unified trade community. If these two nodes belong to the same trade community, then $\delta \left(c_{i},c_{j}\right)=1$, otherwise $\delta \left(c_{i},c_{j}\right)=0.$

## 3. Results

### 3.1. Network Characteristics and Analysis

#### 3.1.1. Network Scale

It is obvious that the global *FTN* is increasingly complex and countries are more interdependent (Figure 1). From 1992 to 2018, the amount of participants increased from 143 to 189. The links in the global *FTN* saw a noticeable upward trend, from 1380 to 3951, a nearly three-fold increase. There are more and more countries participating in food trade, and the structure of the trade network is becoming more complicated.

#### 3.1.2. Network Connectivity

The global FTN has become tighter and more efficient. It is clear from Figure 2 that the figures for network density and *ACC* increased from 0.068 to 0.111 and from 0.33 to 0.479, respectively, over the nearly 30-year period (Figure 2). The figure of *APL* remained stable at just under 2.15, which means that the distance of international trade was short and the transmission efficiency was high. However, *APL* increased significantly in 2002 and 2008, which were due to the increase in trade barriers under the economic crisis and its negative effect on network efficiency.

The power law distribution of the global *FTN* is obvious, showing a scale-free characteristic. According to the degree of the distribution correlation value, the cumulative degree distribution of the global *FTN* was fitted with a power function [33]. We can see from Figure 3 that the degree of distribution of nodes conforms to the characteristics of a scale-free network. In fact, the scale-free characteristic of a complex network is a kind of heterogeneity. The heterogeneity of the global *FTN* was consistently noticeable between 1994 and 2018. In 2018, the top 20 countries in terms of node intensity accounted for about 50% of all trade links of participating countries. The scale-free property of the global *FTN* shows that trade between the node degree distribution is very uneven. As a result, in a scale-free network, a few countries with high nodes occupy the core position of the network and can easily affect the connectivity and stability of the network.

### 3.2. Community Structure and Analysis

#### 3.2.1. Evolution of Community Structure

In the evolution of the global *FTN*, the nodes are relatively fixed and trade links are intensively regionalized, forming a remarkable pattern of clusters. Analysis of the community structure shows that the figure for modularity saw a fluctuating upward trend, from 0.292 to 0.433, between 1992 and 2018. The results indicate that the separation degree was getting larger and the formation of clusters was becoming clearer (Figure 4). In 2010, the degree of modularity reached its peak at 0.479, which may be an effect of the economic crisis. The economic crisis slowed down trade globalization and increased trade regionalization.

From 1992 to 2018, the number of communities remained stable at 4–7 (Table 1). According to the distribution proportion of the members in the communities, the size of the community is uneven. There are communities with a large number of members and clusters with a small number of members (Figure 4).

We choose the years 1994, 2000, 2006, 2012, and 2018 to analyze cluster patterns. Since the end of the Cold War, the global political landscape has evolved from “bipolar” opposition to multi-polarization; economic globalization has accelerated in an all-round way, and regional integration has also accelerated. In 1994, the World Trade Organization was established, promoting further development of economic globalization. The European Union, the North American Free Trade Area, the Association of Southeast Asian Nations, and other regional organizations began to establish or strengthen cooperation. Therefore, the cluster pattern was observed at a time node every six years as of 1994 (Table 2). The trade flows of different communities in different time are shown in Figure 5. In the study, the name of the core country with the largest trade quantity was selected to name the corresponding communities. If there are two countries in a community with relatively comparable quantities of trade, the names of the two countries are referred to together.

In general, global food trade has evolved from “unipolar” to “multipolar” over the past three decades. The original large communities appear to be divided and the internal knot gradually loosing, resulting in a decline of control by traditional powers. The trade gap between different communities has narrowed. Discrete and unbalanced global grain trade pattern that was dominated by the United States gradually changed into a balanced pattern with the United States, Russia, India, France, and other centers. The amounts of sub-node countries continued to increase and block trade increased greatly. Trade is mainly distributed in the Americas, Europe, East Asia, and Southeast Asian countries, but the coverage continues to expand. From 1994 to 2018, the global food trade community pattern was relatively stable on the whole. The United States has always been in the core position of the network, and the communities it dominates have always been those with the largest trade quantities. European countries have formed frequent and stable trade relations with each other due to their concentrated geographical location, trade traditions, and similar institutional backgrounds. With the extensive participation of Southeast Asian countries and South American countries, communities dominated by developed countries has gradually declined their trade quantities. As the main grain producers, Brazil and Argentina have formed trans-continental trade blocks with many countries, and built extensive connections with Asian, Oceanic and African countries. The rising status of Asian countries in the global *FTN* changed the pattern that was focused on the Americas and European countries. Asia has also formed a relatively clear agglomeration group, and has more trade contacts with African countries. Notably, there is a significant feature of the changes in community membership with African countries increasingly participating in each community.

#### 3.2.2. Community Pattern in 2018

In order to further clarify the cluster distribution characteristics of the grain trade, this paper conducts a detailed analysis of the global *FTN* in 2018. The global *FTN* formed six communities that were dominated by major food-exporting countries (Figure 6). The internal trading structure of different communities are shown in Figure 7. Listed as follows: (1) the USA and its importers, (2) Ukraine and Spain and its importers, (3) Russia and its importers, (4) Argentina and France and its importers, (5) Thailand and India and its importers, and (6) Brazil and Iran and its importers.

USA community: This community with the largest trade quantity is America and its importers, which includes 35 countries or regions, mainly distributed in North America, East Asia, and Central America, accounting for about 40% of trade quantity among all countries. America is the most important node in this community, which has rich and large export trade relations with sub-center members, such as Japan, Morocco, and Canada. The community presents a single core network structure.

UKR–ESP community: The second largest community is the Ukraine–Germany community, which consists of 62 members, mainly distributed in Europe and Asia, accounting for about 28% of the trade quantity of all countries. Ukraine occupies the core position, and its trade partners are relatively equal with no obvious secondary centers. Most members of the community have bilateral trade relations with each other and internal trade is extremely intensive. Part of the trade flow reflects the radiation effect of the Ukraine on Asian countries (regions).

RUS community: The Russian community is the third largest community, which consists of 36 members and accounts for about 11% of the global trade quantity. The community is characterized by central-subcenter structure. Russia is the largest food exporter that trades with all members of the community. Egypt, the secondary center, is the largest food importer in the community, with a strong import trade dependence on Russia. Other marginal members are mostly from Africa and Asia, and their trade quantities vary less.

ARG–FRA community: The Argentina–France community, the fourth largest, consists of 15 members, accounting for about 10% of the global trade quantity. This community is more inclined to trade between countries within the same region, and presents a multi-core network structure, with Argentina, France, and Algeria as the core nodes inside the community. There are abundant large trades between the core nodes, which have typical rich-club and negative matching characteristics. Algeria is the largest importer in this community and all its food imports come from the two core members, Argentina and France.

THA–IND community: The Thailand–India community, the fifth largest community, includes 31 members, mainly from Africa, Oceania, and Southeast Asia, accounting for about 5% of the global trade quantity. The community has a dual core structure with Thailand and India as the main core, and Australia and the Philippines as the secondary core. The core nodes have high agricultural production capacity. Their main trade objects are not limited to neighboring regions, but extend their radiation scope to areas with high demand for food around the world.

BRA–IRN community: The sixth largest community, the Brazil–Iran community, includes 10 members, accounting for about 5% of the global trade quantity. Brazil and Iran occupy the core position of this community with absolute large trade flows. The large trade relationship between the two core members is significant and the concentration of community trade is high. Brazil exports to all members in the community and holds a monopoly on food exports. The community has obvious characteristics of network ordering and heterodistribution. All the countries have trade relations with the core country, Brazil, but the relation with other countries is sparse.

### 3.3. Positions of Core Countries

Over the nearly 30-year period, the core trading countries were relatively fixed, but their trade statuses fluctuated to a certain extent. The top 10 out-degree countries include the USA and Canada in North America Argentina and Brazil in South America, France and Italy in Europe, and Thailand, China, India, and Pakistan in Asia. These countries have a stronger control and greater comparative advantage in terms of food export trade. By establishing trade links with more countries, they have diversified markets and enhanced their ability to avoid risk. The top 10 in-degree countries are USA and Canada in North America, and the UK, the Netherlands, Belgium, Germany, France, Italy and Spain in Europe, and South Africa (Table 3). These countries have a wide range of trade partners, so they can prevent trade risks by dispersing the source countries of imports and reducing their import dependence on one country. It is worth noting that the USA always has a central position in the import and export trade network, indicating its strong ability to control world food trade.

The trade network is highly concentrated, and the major importers and exporters of food are dislocated in geographical space. As shown in Table 4, the top 10 countries account for 80% of global food exports. In 1992, the top 10 countries of weighted out-degree were the USA and Canada, in North America, Argentina, in South America, France and Britain, in Western Europe, Thailand and China, in Asia, etc. In 2018, the spatial distribution of trade quantity tended to be balanced, with North America, South America, Europe, Asia, and Oceania divided equally. The center of exports shifted to South America. Europe’s traditional export powerhouses have been replaced by upstarts, such as Russia and Ukraine. From 1992 to 2018, the spatial distribution of trade imports was stable with changes. In 1992, the countries with a high in-degree were mainly Russia, Italy and Holland in Europe, Japan and South Korea in Asia, Brazil in South America and Egypt and Algeria in Africa. In 2018, Mexico became the largest grain importer, while other big importers did not change much. The in-degree of some big importing countries is relatively low, indicating that they have high import dependence, so the uncertainty of external food supply leads to higher food security risks.

## 4. Discussion

With the development of economic globalization, food trade has become wider and tighter. Food trades break resource restrictions and redistribute agricultural resources of national grain production, which play an important role in adjusting food varieties, diversifying supply sources, and obtaining high-quality farm products [34]. In addition, food trade enable the flow of dietary nutrients between countries, altering the nutrient supply of countries and meeting the diversified nutritional needs of different people [35]. Free and smooth food trade reduces the cost and economic burdens of healthy diets, which improves food availability and nutritional diversity. To some extent, it is a great way to promote food security.

However, the global food trade is still controlled by a few core countries, such the USA, Japan, Russia and Brazil. International trade has increased the complexity of the global food system and may increase a country’s exposure to external disturbances. A community structure with a greater intensity of internal cooperation and competition that is formed by core countries is pushing the world trade network toward a “robust yet fragile” configuration. In this network structure, trade networks are more vulnerable when core exporters restrict exports during periods of global food market stress. Political instability, natural disasters, and public emergencies in core countries may affect the security of their external food supply through trade [21]. For example, the sudden outbreak of COVID-19 in 2020 triggered a global recession and a contraction in food trade. Some big exporters, including 14 countries, such as Russia, suspended or banned grain exports, which disrupted global food supply chains. As a result, it increased the vulnerability of some countries’ food systems.

There are three measures can be taken to ensure global food security. First, to protect and optimize existing trade patterns. We should pay a great deal of attention to key hubs in the global FTN, such as the United States, Russia, India, etc. Calling on these trading powers to reduce trade restrictions and keep the global food trading network mobile. At the same time, promoting the multi-polar development of grain trade. Promoting the participation of more countries in food trade can further improve a country’s participation in small group trade, such as in the Thailand–India community and Brazil–Iran community in the future, and promote multi-polarization of the trade network. Spain, South Africa, and other countries with high dependence and concentrations on food imports should further expand trade links with other countries to disperse food trade risks and ensure food supply. Third, we should foster a global vision of a community with a shared future and actively participate in global governance and institutional reform for food security. At present, all countries are in a complex food trade system, with trade agglomeration and increasing dependence. Therefore, in the face of complex domestic and international food markets, countries should strengthen international cooperation, enhance collective action capacity, jointly build a coordinated global food security policy, and ensure food security and stability.

## 5. Conclusions

Trade plays an important role in the global food system. Over the past 30 years, food trade has expanded and the global *FTN* has become more complex. The global *FTN* has the feature of being scale-free, which means that a few countries occupy a large number of trade flows and hold the main share of global food trade.

From 1992 to 2018, the global food trade communities were stable and the number of communities fluctuated from 4 to 7. Trade agglomerations formed by the American community and the French community are relatively stable. While communities dominated by the Ukraine and Russia, respectively, were consistently increasing their radiation capacity in the trade pattern. The leading countries in the network are mainly distributed in North America, South America, Europe, and Asia, with a certain spatial continuity in the continental scope. African countries are involved in different groupings. The cluster pattern reflects the geographical distribution pattern of global food production resources and the geographical proximity of closely connected countries.

Over the nearly 30-year period, the core trading countries were relatively fixed, but their trade statuses fluctuated to a certain extent. The core exporting countries are stable and concentrated, while the core importing countries are relatively dispersed. The nodes with extensive trade relations are concentrated in Europe, North America, South America, East Asia, Southeast Asia, and South Asia. The nodes with large trade quantities are concentrated in the Americas, Europe, and Asia, which largely coincide with the distribution of major food-producing areas.

## Figures and Tables

**Figure 1 foods-10-02657-f001:**
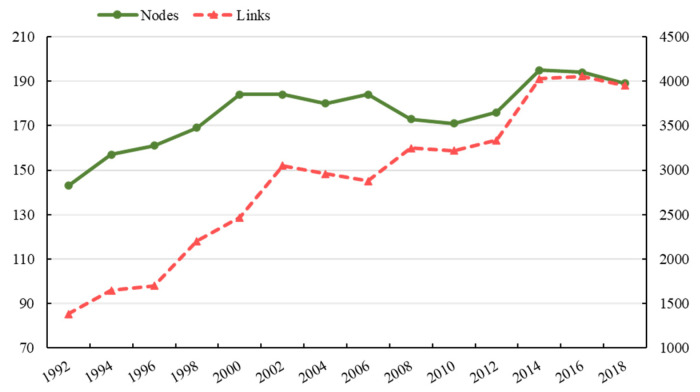
The evolution of the scale of the global FTN.

**Figure 2 foods-10-02657-f002:**
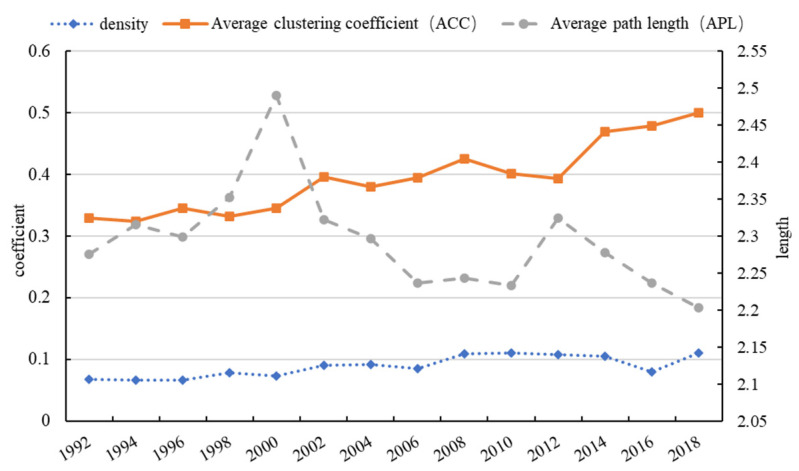
The evolution of density, average clustering coefficient, and average path length of the cereal trade network.

**Figure 3 foods-10-02657-f003:**
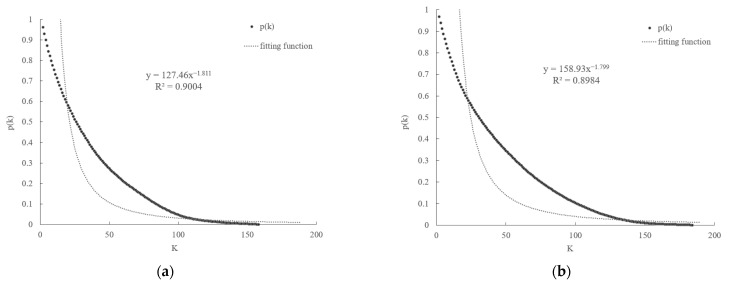
The accumulative distribution of the global *FTN*. The distribution curves for different years are as follows: (**a**) in 1994; (**b**) in 2000; (**c**) in 2006; (**d**) in 2012; and (**e**) in 2018. X represents the degree of node (*k*) in the global *FTN*. Y represents the value of the power law index in degrees. R^2^ expresses the explanatory degree of the relationship between the x and y variables. The values of R^2^ range from 0 to 1. The higher the value, the closer relationship between the x and y variables.

**Figure 4 foods-10-02657-f004:**
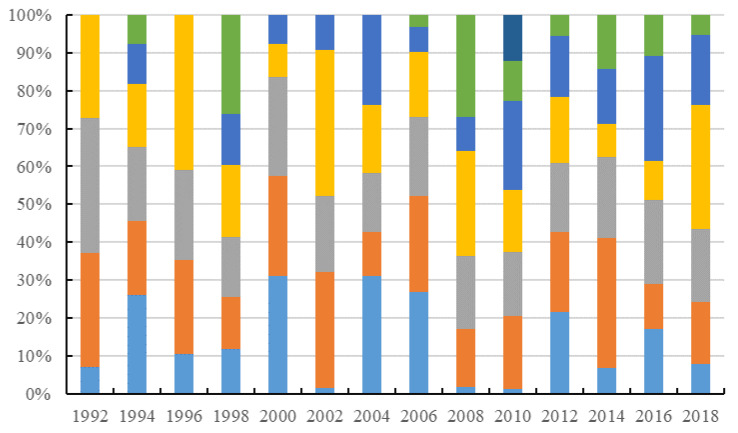
Ratio of the number of community members from 1992 to 2018.

**Figure 5 foods-10-02657-f005:**
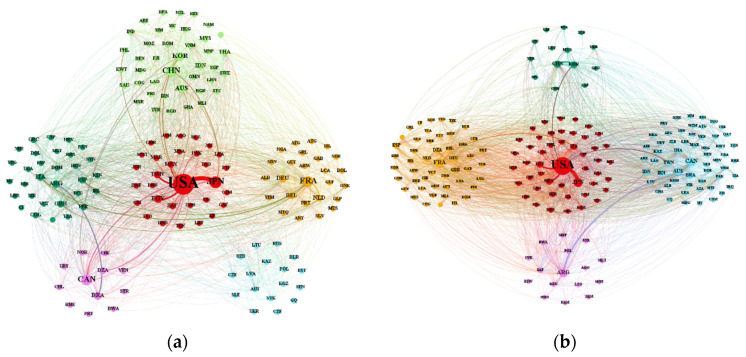
Community structure and flow patterns in the global food trade. (**a**) 1994; (**b**) 2000; (**c**) 2006; (**d**) 2012; and (**e**) 2018. The same color indicates the same trade community. The thicker and larger the lines and dots, the greater the scale of export quantity. The name of the country is briefly processed to facilitate the identification and recognition of the country. In (**a**), red is the USA-JPN community, light green is the CHN-KOR community, yellow is the FRA community, purple is the CAN community, blue is the AUT community, dark green is the ARG community. In (**b**), red one the USA community, blue is the CAN-AUS community, yellow is the FRA community, purple is the ARG community, dark green is the CHN-KOR community. In (**c**), red is the USA community, yellow is the FRA community, blue is the ARG–AUS community, purple is the RUS community, light green is the BRA-ESP community, pink is the ISR community. In (**d**), red is the USA community, purple is the FRA community, dark green is the ARG-BRA community, blue is the UKR-ESP community, light green is the AUS community, yellow is the ISR-CHE community. In (**e**), red one the USA-JPN community, yellow is the UKR-ESP community, purple is the RUS community, blue is the ARG-FRA community, dark green is the THA-IND community, light green is the BRA-IRN community.

**Figure 6 foods-10-02657-f006:**
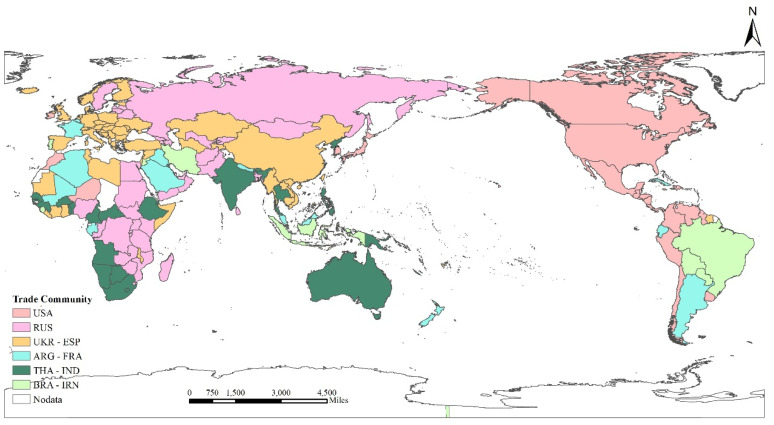
Spatial distribution of global *FTN* communities in 2018. The same color indicates the same trade community.

**Figure 7 foods-10-02657-f007:**
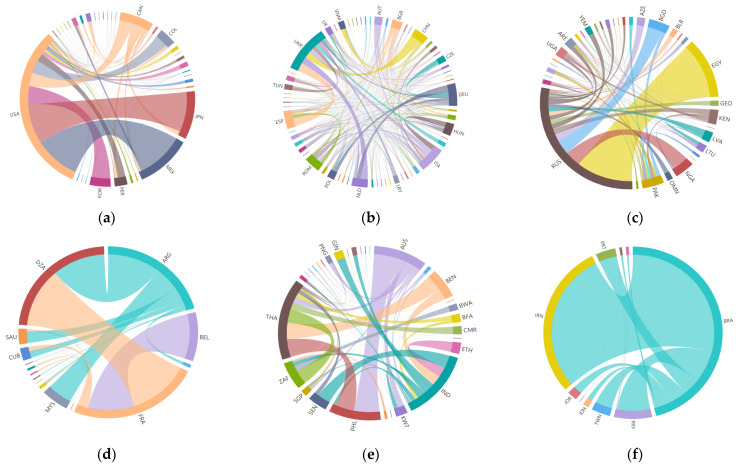
Internal structure of different communities of the global *FTN* in 2018. (**a**) The USA community; (**b**) UKR–ESP community; (**c**) RUS community; (**d**) ARG–FRA community; (**e**) THA–IND community; (**f**) BRA–IRN community. The thicker the connection line, the greater the volume of trade between the two countries and the closer the trade connection.

**Table 1 foods-10-02657-t001:** The evolution of modularity and community numbers.

	1992	1994	1996	1998	2000	2002	2004
Number of communities	4	6	4	6	5	5	5
Modularity	0.292	0.371	0.38	0.396	0.375	0.391	0.407
	**2006**	**2008**	**2010**	**2012**	**2014**	**2016**	**2018**
Number of communities	6	7	7	6	6	6	6
Modularity	0.439	0.476	0.479	0.440	0.419	0.431	0.433

**Table 2 foods-10-02657-t002:** Community detection in 1994, 2000, 2006, 2012, and 2018.

Community	1994	2000	2006	2012	2018
1	USA-JPN	USA	USA	USA	USA
2	CHN-KOR	CAN-AUS	FRA	FRA	UKR-ESP
3	FRA	FRA	ARG-AUS	ARG-BRA	RUS
4	CAN	ARG	RUS	UKR-ESP	ARG-FRA
5	AUT	CHN-KOR	BRA-ESP	AUS	THA-IND
6	ARG		ISR	ISR-CHE	BRA-IRN

**Table 3 foods-10-02657-t003:** Top countries by degrees from 1992 to 2018.

	1992				2018			
	Country	Out-Degree	Country	In-Degree	Country	Out-Degree	Country	In-Degree
1	the USA	84	Netherlands	39	The USA	143	Netherlands	77
2	Thailand	61	the UK	33	Thailand	140	Germany	71
3	Canada	49	France	31	India	132	Canada	68
4	France	48	Belgium	31	Pakistan	120	the UK	68
5	Australia	48	the USA	29	Argentina	120	France	68
6	Italy	44	Saudi Arabia	29	Italy	113	Italy	63
7	Argentina	39	Austria	28	France	106	The USA	61
8	China	35	Sweden	26	Russia	99	Belgium	56
9	Pakistan	33	Denmark	25	China	96	Spain	55
10	Netherlands	32	Italy	24	Brazil	94	South Africa	54

**Table 4 foods-10-02657-t004:** Top countries by weighted degrees from 1992 to 2018.

	1992				2018			
	Country	Weighted Out-Degree	Country	Weighted In-Degree	Country	Weighted Out-Degree	Country	Weighted In-Degree
1	the USA	47,365,973	USSR	21,194,194	the USA	85,014,708	Mexico	22,752,690
2	France	15,884,247	Japan	20,293,443	Russia	36,925,204	Japan	22,173,202
3	Canada	12,251,421	Italy	6,274,973	Ukraine	35,667,668	Spain	15,710,231
4	Australia	9,630,680	Korea	6,174,717	Argentina	30,077,814	Egypt	15,540,185
5	Argentina	7,534,480	Brazil	6,093,616	France	24,062,902	Korea	14,416,776
6	Thailand	5,443,439	Egypt	4,523,293	Canada	23,125,850	Italy	13,371,764
7	China	5,301,685	China, Taiwan	3,842,962	Brazil	21,038,219	Indonesia	13,087,265
8	the UK	2,910,384	Algeria	3,579,009	India	11,088,817	Algeria	12,690,435
9	South Africa	1,798,809	The UK	3,388,189	Australia	10,675,764	Netherlands	11,930,870
10	USSR	1,446,791	Netherlands	3,271,017	Thailand	9,346,882	Iran	10,577,211

## Data Availability

Data available in a publicly accessible repository.

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
