# Peer review of "Evolution of Global Food Trade Patterns and Its Implications for Food Security Based on Complex Network Analysis"

_foods, 2021, doi:10.3390/foods10112657_

Round 1
Reviewer 1 Report
This paper represents an interesting contribution to knowledge around global cereal trade. The paper is sufficiently worked through and presented, yet some aspects should be improved. First and foremost, the methodological choices are poorly justified, especially the formulas for calculating the network measures. The choices of years studied also warrant more elaboration. The framework of “polarization” needs more explanation for the general audience. The discussion is too short and generic for such an empirical underlay and should connect more closely to the results.
Author Response
Dear Reviewer,
Thanks very much for taking the time to review this manuscript. We appreciate all your generous comments and suggestions. Their suggestions have enabled us to improve our work. Based on the instructions provided in your letter, we uploaded the file of the revised manuscript. Please find my revisions in the re-submitted files.
Best regards,
Authors
Response to comments:
Thank you for your summary. We appreciate your recognition of our manuscript. We have revised the manuscript accordingly. Our point-by-point responses are detailed below.
- This paper represents an interesting contribution to knowledge around global cereal trade. The paper is sufficiently worked through and presented, yet some aspects should be improved. First and foremost, the methodological choices are poorly justified, especially the formulas for calculating the network measures.
- Thank you for your careful review. Firstly, we added the reason for choosing the complex network and optimize the expression of model. It can be seen from line 90 to line 116. The contents are as follow: The complex network model can clearly observe the interaction between nodes and the structural characteristics of the cooperative network[30]. The global food trade network is a complex system composed of “nodes(trading countries/regions)”, “edges(trade links)” and “edge weights(billateral trade volume)”,which shows the characteristics of complex network[31]. Hence, the complex network model is conducive to its visual representation of multiple actors in food trade patterns worldwide, which can provide insights into the underlying structure of trade flows, identify key factors within the network whose behaviors may have strong influences on the remainder of the network and judge other descriptive factors. In this case, the global trade network is constructed by taking food trading countries as nodes and food trade links between countries as edges, introducing trade flow to represent as nodes. Global FTN can be abstracted as a weighted complex network G=(N, E, W). N is a node set composed of food trading countries. E is the edge set of food trade relations between countries. W is the set of functions of the trade quantity relationship between two countries. denotes the trade quantity between node and node , i.e. the weight of edges. When , it means that there is no food trade relationship between nation and nation j and when , it means that there are trade flows between nation and nation . The network can be expressed by topological adjacency matrix A=, i.e.
|
(3) |
(4) (1) |
Next, we improve the emphasis of formula for the network measures. The section 2.3 Network measures has been revised.
- The choices of years studied also warrant more elaboration.
We are grateful for the suggestion. Firstly, we add the reason for choose years 1992-2018 to study. It can be seen in line 84 to 82. The contents are as follow: In 1992, the Soviet Union disintegrated and the Cold War ended. China has established a socialist market economy in this year and further integrated itself into the world trading system. The world political and economic pattern changed significantly in 1992. Hence, we selected the annual import quantity data of cereal commodity for 190 countries from 1992 to 2018. Secondly, we unify the study period in section 3.1.2 and section 3.1.2. We choose the year 1994, 2000, 2006, 2012 and 2018 to analyze the network connectivity and cluster pattern.
- The framework of “polarization” needs more explanation for the general audience.
Thank you for your careful review. We add the explanation for “polarization”. It can be seen in line 265-269. The contents are as follow: The original large communities appear to be divided and its internal knot gradually loosing, resulting in the decline of control of traditional powers. The trade gap between different communities has narrowed. The discrete and unbalanced global grain trade pattern dominated by the United States gradually changed into a balanced pattern with the Unit-ed States, Russia, India, France and other centers.
- The discussion is too short and generic for such an empirical underlay and should connect more closely to the results.
We are very grateful for your comments. We revise the part of discussion. The contents are as follow: First, protect and optimize existing trade patterns. We should pay much attention to the key hubs in the global FTN, such as the United States, Russia, India, etc. Calling on these trading powers to reduce trade restrictions and keep the global food trading network mo-bile. At the same time, promote the multi-polar development of grain trade. Promoting the participation of more countries in food trade can further improve the country's participa-tion in small group trade such as Thai-India community and Brazil-Iran community in the future, and promote the multi-polarization of the trade network. Spain, South Africa and other countries with high dependence and concentration on food imports should further expand trade links with other countries to disperse food trade risks and ensure food supply. Third, we should foster a global vision of a community with a shared future and actively participate in global governance and institutional reform of food security. At present, all countries are in a complex food trade system, with trade agglomeration and increasing dependence. Therefore, in the face of complex domestic and international food markets, countries should strengthen international cooperation, enhance collective action capacity, jointly build a coordinated global food security policy, to ensure food security and stability.
Reviewer 2 Report
I found the study to be interesting and potentially very useful, particularly for those who have an academic interest but is not experts in international a trade. The analysis provides a quantitative assessment of highly complex trade patterns that may either confirm or challenge subjective assessments made by experienced trade experts. However, most of the figures do not include legends which identify the different lines on graphs or different colors on charts. The reader should be able to readily associate the different lines and colors with terms that are explained more fully in the text. Text, figures, and tables should complement each other by providing useful information can be understood without relying completely on the other. Also, there are three "figure 2s" in the in the paper, two of which are misplaced and follow figure 5. These problems must corrected.
Author Response
Dear Reviewer,
Thanks very much for taking the time to review this manuscript. We appreciate all your generous comments and suggestions. Their suggestions have enabled us to improve our work.
We are very grateful for your comments on the manuscript. According to your advice, we amended the relevant part of the manuscript. All of your questions were answered one by one.
- I found the study to be interesting and potentially very useful, particularly for those who have an academic interest but is not experts in international a trade. The analysis provides a quantitative assessment of highly complex trade patterns that may either confirm or challenge subjective assessments made by experienced trade experts. However, most of the figures do not include legends which identify the different lines on graphs or different colors on charts. The reader should be able to readily associate the different lines and colors with terms that are explained more fully in the text. Text, figures, and tables should complement each other by providing useful information can be understood without relying completely on the other.
Thanks very much for taking the time to review this manuscript. We add the legends in figure1, figure 3 and figure 5 to identify different lines and colors.
- Also, there are three "figure 2s" in the in the paper, two of which are misplaced and follow figure 5. These problems must corrected.
Thank you for your careful review. We are very sorry for the mistakes in this manuscript and the inconvenience they caused in your reading. We revise the Figure 2. “Spatial distribution of global FTN communities in 2018. The same color indicates the same trade community” into Figure 6.
We appreciate your efforts in reviewing our manuscript during this unprecedented and challenging time. We wish good health to you and your family. Your careful review has helped to make our study clearer and more comprehensive.
Best regards,
Authors